# Shortcomings of Phylogenetic Studies on Recent Radiated Insular Groups: A Meta-Analysis Using Cabo Verde Biodiversity

**DOI:** 10.3390/ijms20112782

**Published:** 2019-06-06

**Authors:** Maria M. Romeiras, Ana Rita Pena, Tiago Menezes, Raquel Vasconcelos, Filipa Monteiro, Octávio S. Paulo, Mónica Moura

**Affiliations:** 1LEAF, Linking Landscape, Environment, Agriculture and Food, Instituto Superior de Agronomia, Universidade de Lisboa, 1349-017 Lisbon, Portugal; fimonteiro@fc.ul.pt; 2Centre for Ecology, Evolution and Environmental Changes (cE3c), Faculdade de Ciências, Universidade de Lisboa, 1749-016 Lisbon, Portugal; anarita.pena@outlook.com (A.R.P.); ofpaulo@fc.ul.pt (O.S.P.); 3CIBIO, Research Centre in Biodiversity and Genetic Resources, Azores Group, InBIO Associate Laboratory, Universidade dos Açores, 9501-855 Ponta Delgada, Azores, Portugal; tiagombm@gmail.com (T.M.); monica.mt.moura@uac.pt (M.M.); 4CIBIO, Research Centre in Biodiversity and Genetic Resources, InBIO Associate Laboratory, Universidade do Porto, 4485-661 Vairão, Portugal; raquel.vasconcelos@cibio.up.pt

**Keywords:** Macaronesia, endemic species, evolution, phylogenetic inference, divergence times, bioinformatics, phylogenomics

## Abstract

Over the previous decades, numerous studies focused on how oceanic islands have contributed to determine the phylogenetic relationships and times of origin and diversification of different endemic lineages. The Macaronesian Islands (i.e., Azores, Madeira, Selvagens, Canaries, and Cabo Verde), harbour biotas with exceptionally high levels of endemism. Within the region, the vascular plants and reptiles constitute two of the most important radiations. In this study we compare relevant published phylogenetic data and diversification rates retrieved within Cabo Verde endemic lineages and discuss the importance of choosing appropriate phylogeny-based methods to investigate diversification dynamics on islands. From this selective literature-based review, we summarize the software packages used in Macaronesian studies and discuss their adequacy considering the published data to obtain well-supported phylogenies in the target groups. We further debate the importance of Next Generation Sequencing (NGS), to investigate the evolutionary processes of diversification in the Macaronesian Islands. Analysis of genomic data provides phylogenetic resolution for rapidly evolving species radiations, suggesting a great potential to improve the phylogenetic signal and divergence time estimates in insular lineages. The most important Macaronesian reptile radiations provide good case-studies to compare classical phylogenetic methods with new tools, such as phylogenomics, revealing a high value for research on this hotspot area.

## 1. Background

During the last two decades, one of the fields that took the most benefit of the explosive growth of bioinformatic tools was phylogenetics, which uses molecular data to establish evolutionary relationships among species [1,2]. Presently, large amounts of genetic data are easily retrievable from public repositories, such as GenBank [3], which stores and organizes molecular sequences, while also making them publicly accessible and ready to use. Other biology focused databases include EMBL (European Molecular Biology Laboratory) [4], UniProt (Universal Protein resource) [5], DDBJ (DNA Data Bank of Japan) [6] and many others, all of them available online, allowing an ever-growing global network of information that continues to expand, giving researchers an almost never-ending flow of new data. However, an issue that arose from the high output of data was its processing. The past two decades have been astonishing, with more than a million-fold improvement in the rate of sequence generation [7], particularly with the introduction of Next Generation Sequencing (NGS), which is further discussed below. With such an extraordinary quantity of sequences being created, the number of bioinformatics tools required to handle such high levels of information, also grew and their analytic capacity dramatically improved. Currently, almost all systematic studies have some type of bioinformatics analysis and it is common to make use of multiple options.

An outstanding area for studies of evolution and speciation is the Macaronesian Region (i.e., Azores, Madeira, Selvages, Canaries, and Cabo Verde archipelagos; Figure 1A) that encompasses a great number of endemic lineages. These islands belong to the Mediterranean Basin biodiversity hotspot [8], which is characterized by high levels of endemism, and represents an excellent area to study evolution. The aim of this study is to review the present knowledge on the tree of life and evolution of Cabo Verde biodiversity using, as case-studies, the reptiles [9] and the vascular plants [10], which show high levels of endemism, with several range-restricted species occurring in different habitats, and thus constitute a good model system to understand patterns of diversification within this archipelago. We will debate the relevance of cutting-edge bioinformatics tools and methodologies to provide a better understanding of the evolutionary history of the recent radiated groups of the Macaronesian Region.

## 2. Phylogenetic Inference and Divergence Time Estimation

The number of bioinformatics tools available today represents a huge framework for each question that arises, presenting several alternative solutions. For phylogenetic analysis, this can prove to be an extraordinary time, as new knowledge, powerful software and hardware frameworks, are continuously improved [11,12]. Three major types of phylogenetic inference methods have been used to retrieve phylogenetic relationships within and between several Macaronesian lineages, namely, Maximum Parsimony (MP), Maximum Likelihood (ML) and Bayesian Inference (BI). Maximum Parsimony (MP) [13,14] has proven to be useful when dealing with recently diverged species with short branches and was commonly used as a standalone solution in earlier Macaronesian studies to resolve the phylogenetic relationships of some plant lineages (e.g., *Convolvulus* (Convolvulaceae) [15]; *Gonosperminae* (Asteraceae) [16]; *Bencomia* (Rosaceae) [17]; *Pericallis* (Asteraceae) [18]). More recent studies have followed the general trend, and usually couple MP approaches with other methods (i.e., ML and/or BI), providing better and more supported hypotheses, namely in several plant lineages: *Lactuca* (Asteraceae) [19]; *Cistus* (Cistaceae) [20]; *Azorina, Campanula, Trachelium, Lobelia, Musschia,* and *Wahlenbergia* (Campanulaceae) [21]; and endemic reptile genera: *Gallotia* (Lacertidae) [22]; *Hemidactylus* (Gekkonidae) [23].

When working with ML methods that were made popular by Fisher in the early 20th century [24], the objective is to infer an evolutionary tree, taking into consideration an evolutionary model which provides rates of nucleotide substitution. Likewise with MP methods, assessing each tree topology is impossible, thus heuristic search methods are used to obtain the most likely tree. More recently, the implementation of BI methods [25,26] became common, and these methods differ from ML by allowing a priori information to be assigned to the alignment, by estimating true probabilities and by resulting in a set of best, most probable, trees. A priori information defines an initial tree with a specified substitution model (e.g., Markov Chain Monte Carlo (MCMC)), distribution of rates across sites, branch lengths and tree topology [27,28].

Programs such as PAUP* (Phylogenetic Analysis Using Parsimony *and other methods) [29] and MEGA (Molecular Evolutionary Genetics Analysis) [30] have been widely used to conduct analyses with most of the above-mentioned phylogenetic inference methods in studies with reptile and with Macaronesian flora lineages [31,32,33,34]. Phylogenetic studies with Macaronesian reptiles [35,36,37,38,39,40,41,42,43,44,45,46,47,48,49] are summarised in Table 1, with detailed information on molecular markers and software used, analyses conducted and corresponding bibliographic reference. Although PAUP and MEGA allow estimating evolutionary models, an essential step in ML and BI analysis is the use of specific software such as jModelTest2 [50] and PartitionFinder2 [51]. Several of the selected research articles make use of these software, instead of only working with more generally purposed programs. jModelTest (and its predecessor ModelTest) seem to be more widely used, but more recent works seem to prefer using PartitionFinder instead (see more details for each lineage in Table 1).

To specifically perform ML and BI analysis with Macaronesian reptiles (Table 1) and plant lineages [19,20,21,31,32,33,34], two main software packages have also been often used: RAxML [52] and MrBayes [53]. Randomized Axelerated Maximum Likelihood (RAxML), and similar software packages such as PHYML [54] and GARLI [55], make use of ML estimations to perform phylogenetic inference; while MrBayes estimates phylogenies using a variant of the MCMC method called Metropolis-coupled MCMC. Newly generated and well-sampled phylogenies that made use of the ML algorithm implemented in RAxML and the Bayesian inference method in MrBayes, have contributed to establish the evolutionary and biogeography relationships within complex Macaronesian plant lineages, namely on *Limonium*, Plumbaginaceae [56] and on Campanulaceae endemic species [21].

Bayesian approaches also allow obtaining divergence time estimations [57,58], an analysis which provides an important insight into the evolutionary history of extant lineages and has been used to some extent with Macaronesian lineages (more details on published studies are provided below). An extremely important step in divergence time estimation is calibrating the clock (see column–Dating: Table 1). To calibrate internal nodes there are two main forms available, specifically, one that makes use of fossil and biogeographical (which requires some a priori knowledge of the relation between the species), and another that uses substitution rates previously calculated (secondary calibration). However, only reliable sources of secondary calibrations should be used, or a considerable error may be introduced in the analysis, as we further discuss in Section 4. As such, comparing several scenarios obtained with different calibration techniques is generally the best methodology.

Due to the complexity of the divergence time estimation methods, the choice of the software to perform these analyses is paramount. The most popular software used in this type of analysis is BEAST [59] and the more recent BEAST 2 [60], which make use of Bayesian multispecies coalescent methods to generate chronograms. However, working with high volumes of data, namely multi-individual and multilocus data, is a challenge, as it exponentially slows the analysis. To solve this setback a specific package was developed, StarBEAST or *BEAST [61], has shown to be very informative when working with rapid radiations [62]. This is of particular importance when working with insular species, as these systems are often fairly recent and under a considerate evolutionary pressure.

Other used approaches in Macaronesian phylogenetic inference studies, have been the estimation of phylogenetic networks, to test the existence of cryptic events (e.g., hybridization, recombination, horizontal gene transfer, or duplication or loss of genes) and often relate to intraspecific genealogies in the given molecular data set. The phylogenetic networks were used to produce complete community-level phylogenies and can be implemented in several software packages like TCS [63] and SplitsTree4 [64]. Particularly, TCS was used to analyse the genetic diversity for endemic plant lineages of Cabo Verde (i.e., *Globularia amygdalifolia* (Plantaginaceae), *Cynanchum daltonii* (Apocynaceae) and *Umbilicus schmidtii* (Crassulaceae) [65]) and of Azores (i.e., *Ammi, Angelica lignescens, Azorina vidalii, Euphorbia stygiana, Pericallis malvifolia*) [66] and of both archipelagos, i.e., *Lactuca* [19]. These methods also provided further insights into the population structure and genetic diversity of widespread plant lineages within Cabo Verde, Azores, Madeira and Canary Islands, such as the Campanulaceae [21], which highlights the usefulness of this tool for this type of assessment. On the other hand, SplitsTree4 contains methods specialized in the creation of recombination and hybridization networks, and were used in some studies with plants [67], where the obtained network revealed conflicting phylogenetic signals, such as rapid radiation or ancient hybridization events, within five Betoideae genera distributed in the Macaronesian Islands and in the western Mediterranean region.

## 3. Macaronesian Islands as Model Systems in Evolution

Islands can be broadly classified as one of two types: continental or oceanic. Continental islands are fragments of a larger landmass, whereas oceanic islands result from volcanoes rising above the water on or near a mid-ocean ridge. Particularly, oceanic islands, as is the case of the Macaronesian Islands, were the focus of studies that contributed to the understanding of speciation and the origins of biodiversity [68]. The terrestrial endemic lineages of the Macaronesian Islands are often characterized by occupying different habitats; showing striking morphological differences among species; frequent rarity and being present in few and small populations [65]. Over the past decades, research has focused on multiple facets of islands and how they have had a relevant role in the establishment of ecological and evolutionary theories [68,69].

Different patterns of colonization within the Macaronesian Region have been an ongoing subject of debate. Some of the islands within this region are near the mainland, with the closest being Fuerteventura in the Canary Islands, which distances only ca. 95 km west of mainland Africa (Figure 1A). This proximity may have facilitated dispersion and colonization with the influence of oceanic currents (North Atlantic and Canary currents), while north-easterly trade winds may have fostered the dissemination of ancestral species from the continent and nearby Canary Islands. For this reason, it has been proposed that frequent colonization events from the continent and close archipelagos may have contributed to the high number of endemic species in some of the less isolated Macaronesian Islands [70].

## 4. Phylogenetic Resolution and Divergence Time Estimation among Macaronesian Insular Groups–Case-Study Using A Meta-Analysis of Reptile Data

The wide variety of habitats within the Macaronesian Islands is home to a rich biodiversity, and over 1000 endemic plants have been identified so far. The evolutionary history of this huge diversity is not properly understood, and many lineages remain understudied, particularly in Azores and Cabo Verde. In fact, for Cabo Verde, few lineages were subject of molecular dating analyses to date, specifically: *Campanula* [21,71]; *Diplotaxis* [31]; *Echium* [32]; and *Lotus* [72], which represents about one-fourth of the total vascular plant diversity when considering lineages with more than one endemic species in the archipelago. On the other hand, only six reptile genera (i.e., *Chalcides*; *Chioninia*; *Gallotia*; *Hemidactylus*; *Tarentola* and *Teira*) occur within the Macaronesian Region (see Figure 1B), and from these, only three occur in Cabo Verde [9]. Due to the representativeness of reptile genera occurring in Cabo Verde, these taxa were selected as indicators of the present state of lineage resolution and divergence time estimation within Macaronesia. Twenty-three publications dealing with this group were selected, and a meta-analysis was conducted based on the methodology followed and results obtained. The present compilation included data from: (i) gene regions used; (ii) inference methods used to reconstruct phylogenies; (iii) completeness of the phylogenies generated. (iv) methods used to estimate the divergence times within each taxonomic group; (v) divergence times estimated.

The meta-analysis conducted revealed that although a huge number of methods and standard programs were used (Table 1; Figure 2, Figure 3 and Figure 4), there are still important knowledge gaps for some lineages. This analysis supports the idea that most of the endemic lineages are monophyletic but the huge morphologically diversity within some of these lineages was not always fully represented by the analysed taxa. On average, only 67%, 47% and 61% of the existent taxa were analysed in the selected studies for *Chioninia*, *Hemidactylus* and *Tarentola*, respectively (Figure 2). Furthermore, even within the most well-sampled taxonomic groups, outgroup selection was not consensual which might have introduced a bias in the obtained phylogenies.

While most of the divergence time estimations published to date were obtained with the application of robust bioinformatics methods, such as BEAST or *BEAST, it is evident that the lack of fossils for most lineages was an important impediment for the accurate calibration of molecular clocks (Figure 3 and Figure 4; only three studies used fossils). Due this shortcoming, several of the studies made use of mutation rates and/or geological information to date their phylogenies, which can lead to considerable error (Figure 4).

Geological dating in particular, can induce a significant amount of error when calibrating, as the accepted age of a given island or archipelago is often submitted to changes as new research methods and hypothesis are built. Some Cabo Verde Islands for example, have recently been found to be younger than previously thought, yet older island ages have been applied to dating methods thus promoting stochastic errors in divergence time estimations [73,74,75]. This means that studies based on the previously accepted geological ages were using an unfitting baseline for their calibrations, which could have led to incorrect estimations and conclusions. Another common mistake that can be made with this methodology is the incorrect assumption that all islands, and in particular those of volcanic origin and from the same archipelago, share a similar history throughout the eras and are able to harbour life all along their cycles, from start to present [76]. Each island has its own unique characteristics, and undergoes both long and short periods of change, as well as stationary phases, which dramatically changes their conditions and evolutionary pressures [77]. Cabo Verde archipelago is a good example of such a system, with different islands leading to diverse colonization events and endemic lineages. The glacial and interglacial periods are of special importance to the archipelago, since during the glacial periods, these regions were buffered from most of its effects (with temperatures variating very little) due to their tropical geographical and worked as a refuge for several species that eventually became extinct in neighbouring regions [78]. The rise and fall of the sea levels also plays an important part when studying evolutionary processes, and in particular with plants and reptiles, due to their influence on the available terrestrial area of the islands. During the interglacial periods the terrestrial area can increased dramatically (for example, in Cabo Verde, it was essentially doubled during the Last Glacial Maximum), which can allow for land bridges to form between neighbouring islands (e.g., Santa Luzia, Branco and Raso) and allow gene flow processes to occur, leading to a different evolutionary history of these thought-to-be isolated species. On the other hand, during glacial periods, this connection is severed, which not only leads to isolation and possible speciation events but can also induce several extinctions, as some organisms may not adapt to the new conditions. As such, when taking into account geological dates, particularly for oceanic volcanic islands, one must consider how these systems are often found to have very complex and non-linear geological histories. For non-insular systems, the use of this type of calibration is not as frequent, as dated continental geological events are usually punctual and restricted to particular regions [79]. The Qinghai-Tibetan Plateau (QTP) is a good example of a continental feature that has undergone several biological studies that used its formation as a calibration point to date their lineages [80,81]. The QTP and similar geological features are usually well-explored and dated but, like in insular systems, some care is necessary to assure that the calibrations are fitted to be used in a given phylogenetic reconstruction, and if possible, other complementary calibrations should be used.

All these factors hamper the development of well-supported hypotheses for the evolutionary history of the reptiles, a large portion of the endemic diversity found in the Macaronesia Region (see Figure 1B). Nonetheless, the vast amount of knowledge on Macaronesian reptile lineages has been continuously increasing along the years, and with the introduction of more advanced and progressive analysis methods, several knowledge landmarks have been achieved, such as the clarification of the *Chioninia* genus from the Mabuya superfamily [82] or the several phylogenetic studies on the Cabo Verde endemic lineages, that brought a clarification on their evolutionary story [39,49,83]. Recently, other methods opened new perspectives, namely the identification of plants, arthropods and bird species predated by different reptile taxa, using NGS methods (metabarcoding of faecal pellets), and a comparison with reptile lineage diversification occurring in different Cabo Verde Islands [84].

Over the past years, it was possible to witness a progressive development of the methodologies used in systematic studies of Macaronesian groups. Earlier studies used only a molecular method, such as MP, and in the case of plants, only one or two markers were used [15,16,17,18]. During the past decade, the combination of different methods, particularly the ML with BI analyses, and the addition of more DNA markers, allowed the development of more accurate phylogenies. For the Macaronesian reptiles, this new approach has proven to be very useful to reconstruct their evolutionary histories, as well as to clarify their taxonomic position. Possibly the most complex group of the three Cabo Verdean genera, the *Chioninia,* has undergone several taxonomic rearrangements and multiple analyses with sometimes disparate results. Problems with this group were mainly found in their relationship with its sister groups, which were previously assigned to a single large genus, *Mabuya*. Some studies place the *Chioninia* reptiles as close relatives to Middle-Eastern *Trachylepis* and some *Mabuya* species [85,86], while others place them alongside other African and Madagascan lineages [87]. A recent work by Karin et al. [37] was finally able to reconstruct a well-supported phylogeny for this reptile group that placed the *Chioninia* as a sister clade to Afro-Malagasy *Trachylepis*. To achieve this result, a wide sample of reptilian was used, and perhaps most importantly, several markers of both nuclear and mitochondrial loci were obtained, unlike previous works, which only used mitochondrial DNA.

Comparatively, the genera *Hemidactylus* and *Tarentola* are far less intricate, while still presenting their own unique challenges. The *Hemidactylus* genus is known for being one of the largest and more widespread reptile groups, with more than 100 described species, which makes it a common species to study due to its representativeness and multiple colonization events. Therefore, this is the well-studied group of the three Cabo Verde genera, both in the number of studies and its evolutionary and diversification history. Another positive feature of this reptile group is the availability of several fossil records, such as *Sphaerodactylus,* which were used to calibrate several *Hemidactylus* phylogenies [41,42,44,45]. As we previously discussed, fossil calibrations are generally very informative and precise when used properly, i.e., when they are properly justified [88,89]. The combination of these two factors (number of studies and fossil calibrations), has led to a mostly well-known phylogeny, which once again highlights the importance of the development of new methods of analysis and their use in numerous complementary works, which then lead to better and more interesting solutions. 

Out of the three, the *Tarentola* is the most represented genus in the Macaronesia region, with some species in the Canary Islands, Cabo Verde, Madeira and Selvagens, while its origin region is thought to be the American continent. This group can be used as an example on how regionally focused studies can work in par with comprehensive ones that include all the species of a genus. Research in the Cabo Verde *Tarentola* species has been quite thorough, with several studies continuously improving the knowledge of their taxonomy and diversification patters [47,49]. In parallel, other studies shed some light in the evolutionary history of the genus, by hinting at a single colonization event from America to Africa [44], and by clarifying the relations between the several Mediterranean species [48]. When we combine these works, we can find a more robust solution, one that explains how and when this genus originated, how it came to Cabo Verde and how it has evolved since then, which is a collective knowledge that would not at all be possible with just one of these finds.

Even with all challenges considered, reptiles are still one of the most well-studied groups within the Macaronesian region, and their evolutionary patterns appear to be clearer and more comprehensible (Figure 2). Plants on the other hand, have a huge diversity of lineages that only recently have been studied using large-scale sampling within each archipelago and wide sets of molecular markers to trace their phylogenies, as is the case of the Plumbaginaceae [56] and Campanulaceae endemic species [21]. Furthermore, polytomies still persist frequently in plant phylogenies. For instance, the molecular phylogeny recently published by Franzke et al. [31] for the biggest plant radiation of Cabo Verde Islands (i.e., *Diplotaxis,* Brassicaceae) revealed that the tree is largely unresolved, and the clades within the tree show very weak support (55–69%). Therefore, particularly for plants, there is a need to perform more studies to understand the evolutionary patterns and dating analyses [90,91].

## 5. The Potential of Phylogenomics to Understand Evolutionary Relationships in Insular Lineages

As stated above, phylogenetic methods used to date to establish the relationships among insular lineages were mostly based on single locus or a small number of loci (i.e., nuclear and plastid genes for plants and mitochondrial genes for animals) and often the resulting trees included unresolved clades and were thus unable to fully reflect the speciation events within the Macaronesian Region.

Considering (i) the inherent drawbacks of the targeted molecular marker-based techniques, namely, the high effort required for finding a considerable number of informative loci; (ii) their little resolving power at low taxonomic levels, which ignores many population-level effects, and (iii) increasingly affordable sequencing costs resulting from the continuous advancement of next-generation sequencing (NGS) technologies, phylogenomics emerges as an promising alternative tool for studying evolutionary relationships based on comparative analysis of genome-scale data [92].

Phylogenomics can be seen as the phylogenetics branch which specializes in inferring species relationships based on whole genomes (WGS) or genome-partition approaches and is starting to be applied in evolutionary and ecological studies on Macaronesian islands [93,94]. The access to genomic data has proven to potentially alleviate previous problems of phylogenetics that resulted from stochastic error (limitation of sampling few genes) by expanding the number of characters [95]. However, genome wide analysis is not always straightforward, particularly when working with non-model organisms or large and complex genomes. On whole genomes approach, plastid sequencing is one of the most technically accessible regions of the genome, and its sequence conservation makes it a valuable region for comparative genome evolution, phylogenetic analysis and population studies [91]. Recent studies on endemic plants using insular lineages [96], and particularly from Azores archipelago (i.e., *Laurus azorica* (Seub.) Franco, [97]) show some promise to help resolving species relationships at phylogenetic level. From the various genome-partitioning strategies to sequence-selected subsets of the genome (reviewed in [96]), restriction site associated DNA sequencing (RAD-Seq) has emerged as powerful alternative to WGS, where is sequenced a representative portion of the genome, which allows for a greater depth of coverage and less financial costs, compared to complete genome sequencing [98,99,100]. As such, several studies found different uses for this methodology, that range from alignment and gene identification processes, to advanced genomic structuring evaluation [101]. Even though this method is broadly used, there are still issues to be taken in consideration, namely the high bioinformatics hub to detect loci involved in adaptation and to infer possible linkage disequilibrium effects in the data [102].

Another consideration when performing RAD-Seq analyses is the type of assembly to be used–de novo or referenced-based. As complete genomes are still scarcely available for most of the Macaronesian lineages, the selection of the method is usually limited, and the most viable solution is a de novo assembly [103]. The use of de novo assembling may provide interesting results, even when faced with a substantial volume of missing data, due to the higher number of loci on the analyses [104]. 

A scenario of rapid plant and reptile radiations is found among most of the Macaronesian lineages, with molecular differences not being correctly depicted by traditional molecular markers and phylogenetic analyses, mostly due to low accumulation of differences in the genomic regions screened between different species in different islands or even within species at intra-islands levels/different habitats [104,105]. Macaronesian Islands can also be used as model to study evolutionary processes of genotype-to-phenotype adaptations, and explore the adaptation of different taxa to certain habitat conditions, within each archipelago. Thus, when searching for adaptive traits, proper genomic tools are needed to detect population genetic divergence associated with processes of local adaptation [106]. NGS technologies can produce informative molecular markers that could address population differentiation and phylogeography issues, as well as find signatures of selection at a genomic scale [107]. High-throughput discovery of single nucleotide polymorphisms (SNPs) using RAD-seq [104] can also be effectively conducted on species without a reference genome, enabling studies in non-model organisms as was recently published for reptiles from Canary Islands [108]. By assessing diversity in traits under strong selective pressure, population genomics can be used to identify signatures of selection and thus genes of adaptive/fitness significance can be potentially recognized [109]. Also, SNPs that are linked to traits under selection, are highly valuable for identifying genetic loci/regions that contribute to phenotypic variation. As such, the genetic basis of adaptation under an island scenario could be a significant advance in insular evolutionary discipline, for example, could be explored to tackle genetic divergence between different species to assess colonization events, genetic variability among endemic taxa and identification of signatures of selection at a genomic scale linked to adaptive evolution on taxa present in the same environment (i.e., dry conditions) [109,110]. In an insular context, the occurrence of morphologically similar but ecologically diverse species on each island, or even within different habitats of the same island is fairly common, thus making insular ecosystems the perfect models to investigate niche shifts and adaptation [108].

In regard to insular evolutionary studies, RAD-Seq has proven to be very effective to contribute to clarify the relationships within closely related taxa, as well as providing a good alternative to study groups where Sanger sequencing failed to provide accurate results [93,111]. In cases where closely resembling species were involved, it is also generally proficient in providing good phylogenies, even when faced with poor genetic signals and conflict between gene trees [112]. Studies using RAD-Seq technology plants from Macaronesian Islands were able to resolve relationships in lineages that have undergone recent, rapid diversification resulting in extensive ecological and morphological diversity, as in *Tolpis* [95] and *Micromeria* [113]. 

Finally, other phylogenomics methods can be used to address the current shortcomings of relatively deep phylogenetic divergences (several million years) to shallow-scale divergences are based target enrichment to sequence protein-coding exons [114,115] or anchored hybrid enrichment [116]. High-throughput sequencing (HTS) techniques provide the opportunity to use thousands of loci to address phylogenetic problems. As well as these new technical capabilities created by NGS, there are also new types of problems that have to be addressed in phylogenomics, such as incomplete lineage sorting, gene duplication and loss, and horizontal transfer.

## 6. Final Considerations

The interdisciplinary field of bioinformatics was born from the need to develop and employ methodologies to study large amounts of biological data, namely proteins and DNA. Many scientific areas benefited greatly from the introduction of specialized and powerful tools that improved the implementation of databases, data analysis and biological interpretations. For evolutionary study fields such as phylogenetics, the knowledge jump has been substantial since the introduction of bioinformatic methods in their analysis. Making use of highly advanced software packages it is now possible to perform high-throughput DNA sequencing, and thoroughly analyse this data with several methods of phylogenetic inference, to obtain informative and well supported phylogenies. Moreover, the introduction of NGS was one of the biggest achievements in the biology field in the last couple of decades. Several biological fields were revolutionized and some only reached their true potential when high-throughput sequencing methods were finally widely available at a relatively low cost. For phylogenetic sciences, this was an revolutionary occurrence, as it was finally possible to sequence, store and analyse complete genomes, and in doing so, reconstruct more comprehensive phylogenies.

The power of phylogenetic sciences has been crucial as tools in the development and consolidation of knowledge of insular native/endemic species in Macaronesia, clarifying their classifications, understanding genetic patterns and evolutionary processes that occurred on the islands and/or archipelagos. The combined data from phylogeny and divergence time estimations are able to clarify diversification hypothesis and assign probable colonization events, and can even help discover new species, providing useful information for the field of conservation biology in insular environments. Phylogenomics are at a turning point and it is highly likely that in the next few years, new understanding on several issues regarding species diversification and adaptation will be instigated by the application of new methodologies to natural laboratories such as the Macaronesian Islands.

## Figures and Tables

**Figure 1 ijms-20-02782-f001:**
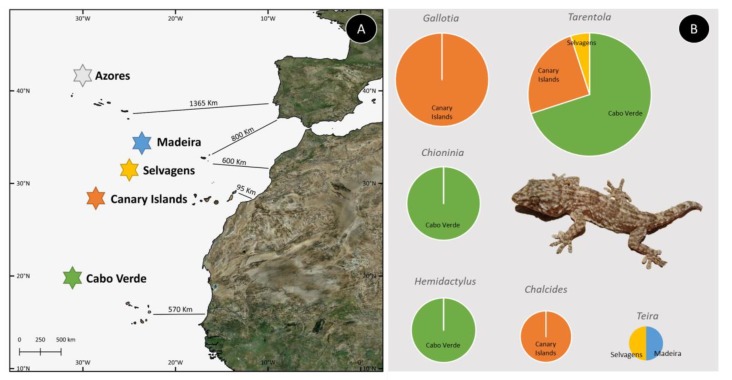
The five archipelagos of the Macaronesian region—Azores, Madeira, Selvagens, Cabo Verde and Canary Islands—with the corresponding distance to the closest mainland point (**A**), and the diversity of endemic Squamata reptile taxa within this region (**B**): *Chalcides (coeruleopunctatus, sexlineatus, simonyi* and *viridanus*), *Chioninia (delalandii, fogoensis, nicolauensis, spinalis boavistensis, spinalis maioensis, spinalis salensis, spinalis santiagoensis, spinalis spinalis, stangeri, vaillanti vaillanti* and *vaillanti xanthotis), Gallotia (atlantica mahoratae, atlantica atlantica, atlantica laurae, auaritae, bravoana, caesaris caesaris, caesaris gomerae, galloti eisentrauti, galloti galloti, galloti insulanagae, galloti palmae, intermedia, simonyi machadoi* and *stehlini*), *Hemidactylus (bouvieri bouvieri, bouvieri* ssp., *bouvieri razoensis, boavistensis* and *lopezjuradoi*), *Tarentola (angustimentalis, bischoffi, boavistensis, bocagei, boettgeri boettgeri, boettgeri hierrensis, caboverdiana caboverdiana, caboverdiana raziana, caboverdiana substituta, darwini, delalandii, gigas gigas, gigas brancoensis, fogoensis, gomerensis, maioensis, nicolauensis, protogigas protogigas, protogigas hartogi* and *rudis),* and *Teira dugesii*. The three extinct taxa (*Chioninia coctei, Gallotia goliath* and *Gallotia simonyi simonyi*) were not included in this representative analysis. Photo of *Tarentola maioensis* by R. Vasconcelos.

**Figure 2 ijms-20-02782-f002:**
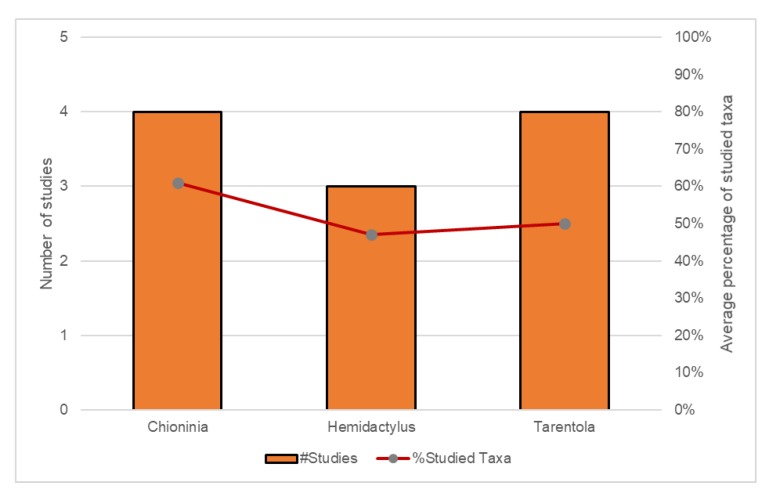
Number of studies (columns) and percentage of studied taxa (red line) within Macaronesian reptile lineages.

**Figure 3 ijms-20-02782-f003:**
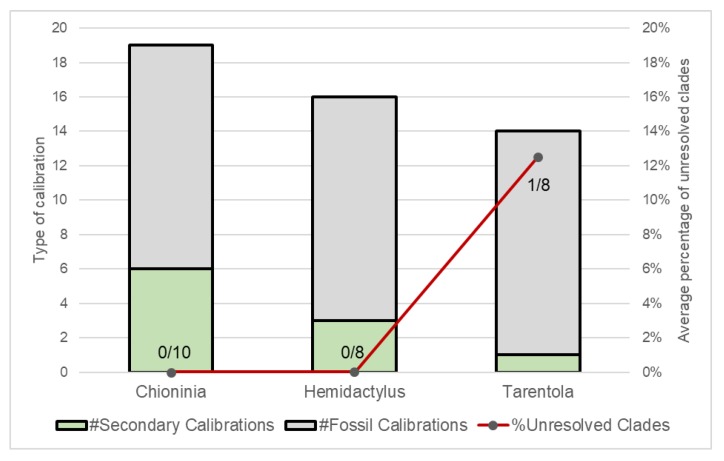
Calibration methods used with Macaronesian reptile lineages. Total number of calibration points (columns) and number of unresolved clades (red line). The average percentage of unresolved clades is shown within the columns as the number of unresolved clades/total number of clades.

**Figure 4 ijms-20-02782-f004:**
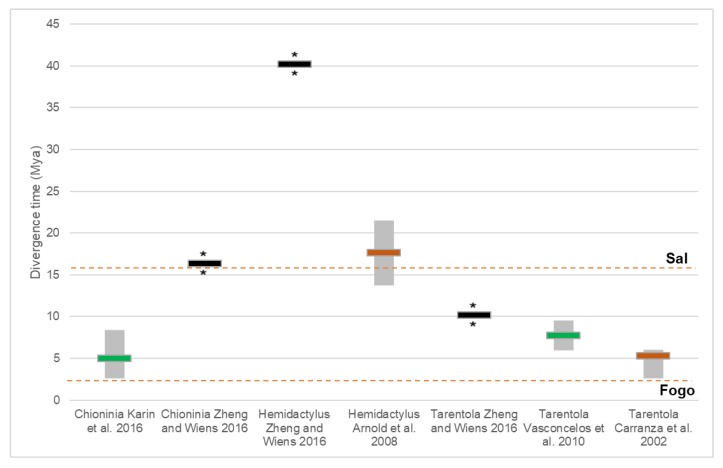
Divergence times obtained within Macaronesia reptile lineages in a meta-analysis of selected studies. Horizontal bars represent average age estimation (Mya), grey bars indicate minimum and maximum ages estimated (Mya). Asterisks indicate lack of data. Horizontal bar colour indicates the calibration method used: green–mutation rate; black–fossil; brown–mutation rate and geological. The age for the Fogo and Sal islands is shown in a dashed orange line (reference for age; [73]).

**Table 1 ijms-20-02782-t001:** Phylogenetic studies with Macaronesian reptiles, with molecular markers and software used, analyses conducted and corresponding bibliographic reference.

Family/Genus	Markers	Analyses	Software	Dating	Year	Ref.
**Scincidae**						
*Chalcides*	cyt b, 12S, 16S	ML+BI	ModelTest, MrBayes, PhyML	Mutation rate (12S, cyt b) + Geological	2008	[35]
*Chalcides*	RAG1, BDNF	BI	MrBayes, BEAST	Mutation rate + Geological	2012	[36]
*Chioninia*	16S, ND2, BDNF, BRCA1, BRCA2, CMOS, EXPH5, KIF24, MC1R, MXRA5, RAG1	ML+BI	PartitionFinder, RAxML, MrBayes	Mutation rate (16S, ND2)	2016	[37]
*Chioninia*	16S, ND2, BDNF, BRCA1, BRCA2, CMOS, EXPH5, KIF24, MC1R, MXRA5, RAG1	BI	BEAST	Mutation rate (previous works)	2016	[38]
*Chioninia*	cyt b, COI, 12S	ML+BI	jModelTest, PhyML, MrBayes	Mutation rate (12S, cyt b)	2011	[39]
**Lacertidae**						
*Gallotia*	cyt b, 12S, 16S	ML, MP	ModelTest, PhyML, PAUP	Mutation rate (12S, cyt b) + Geological	2006	[40]
*Gallotia*	cyt b, 12S, 16S, COI	BI, MP	MrBayes, TNT, BEAST	Geological	2010	[22]
**Gekkonidae**						
*Hemidactylus*	ND2, RAG1, PDC	ML+BI	PartitionFinder, RAxML, MrBayes, BEAST	Fossils	2014	[41]
*Hemidactylus*	Cmos, 12S, RAG1, PDC	ML+BI	ModelTest, Paup, BEAST	Fossils + Mutation rate	2013	[42]
*Hemidactylus*	cyt b, 12S	ML	ModelTest, Paup, PhyML, MrBayes	Mutation rate	2006	[43]
*Hemidactylus*	12S, cyt b, cmos, ND4, MC1R, RAG2	ML+BI	RAxML, MrBayes	Mutation rate (12S, cyt b)	2012	[23]
*Hemidactylus*	RAG1, RAG2, Cmos, ACM4, PDC	ML	RAxML	Fossils	2011	[44]
*Hemidactylus*	12S, ACM4, cmos, RAG1, RAG2, PDC	BI	PartitionFinder, BEAST	Fossils + Geological	2016	[45]
*Hemidactylus*	12S, cyt b, MC1R, cmos, RAG1, RAG2	ML+BI	jModelTest, BEAST, RAxML, MrBayes	Mutation rate (12S, cyt b)	2013	[46]
**Phyllodactylidae**						
*Tarentola*	cyt b, 12S, cmos	ML, MP	Paup	Geological	2002	[47]
*Tarentola*	12S, 16S, PDC, ACM4, MC1R, RAG2	ML+BI	jModelTest, RAxML, MrBayes	Mutation rate	2012	[48]
*Tarentola*	cyt b, 12S	ML+BI	jModelTest, MrBayes, PhyML	Mutation rate (previous works)	2010	[49]

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
