# Peer review of "Shortcomings of Phylogenetic Studies on Recent Radiated Insular Groups: A Meta-Analysis Using Cabo Verde Biodiversity"

_ijms, 2019, doi:10.3390/ijms20112782_

Round 1
Reviewer 1 Report
This paper report an important review regarding phylogenetic studies biodiversity in Cabo Verde Island. It is well written. A fairly high number of recent articles were covered by this review.
Author Response
RESPONSE: We thank the reviewer for their careful reading of the manuscript and their constructive remarks.
Reviewer 2 Report
Review of Manuscript ijms-465936 “Shortcomings of phylogenetic studies on recent radiated insular groups: a meta-analysis using Cabo Verde biodiversity”:
The manuscript by Romeiras et al. is, according to the authors, a review of Macaronesian island phylogenetic studies of plants and reptiles and the shortcomings of those methods for studying insular groups. The authors present a description and review of phylogenetic methods in some reptile genera and provide a list of the markers, methods, and programs used for phylogenetic analysis and divergence time estimation in those papers. The authors then attempt to compare studies of reptile phylogeny in the Macaronesian islands by presenting data on the number of studied taxa in a given group, the number of calibrations for divergence time estimation, and the number of unresolved clades for each genus. They then discuss the relative divergence time estimates for different genera among studies and present one or two possible reasons for these differences. Finally, the authors recommend phylogenomics (and RAD-seq in particular) for resolving such issues.
There are some potentially valuable components to both the goals and implementation of this manuscript. First, it is reasonable to question whether past or current phylogenetic methods have been able to resolve phylogenetic questions in any group, and this question is particularly important when thinking about recently radiated lineages due to possible incomplete lineage sorting or hybridization, among other things. Additionally, Table 1 provides very relevant information for any authors interested in pursuing phylogenetic research on these groups, as it is a succinct and thorough list of information from each study. However, I see substantial problems with this manuscript as written, and I detail some of my main points below.
First and foremost, the authors do not provide satisfying information that directly addresses their stated goal of discussing the adequacy of the given methods for the target groups. The beginning of the manuscript focuses on the different steps in phylogenetic inference, and details alignment and phylogeny inference methods (and the software programs one can use), but only mentions some standard caveats to each method, with no real novel insight into how these caveats are impacting their groups of interest. Furthermore, much of this information, including lists of programs that are available for alignment and phylogeny inference, can be found in any number of reviews or book chapters on the subject.
As a reader, I expected a more thorough look at the phylogenies generated using these methods, and a discussion of how they compared among studies that used similar or different methodologies. The authors attempted to do this in section 4 (starting on line 187), but their analyses were underwhelming. They provide no information on the overall consensus of phylogenies generated in the studies they highlight, with the exception of Figure 3 where the number of unresolved clades is presented for each group. However, not only do I not know which y-axis refers to calibration points v. number of unresolved clades, I do not know how the authors are determining what is an “unresolved clade”. What does that mean in the context of their analyses? And how many clades are present in each of those groups? The authors also do not discuss context for each of these groups or discuss how the different evolutionary history in the three genera may impact phylogeny inference.
The meta-analysis results on divergence time are interesting, but somewhat confusing. It is clear that the type of calibration seems to matter (in some genera more than others), but there is no comparison between geologic dating in insular groups compared to dating in non-insular groups. Since their argument is that islands, particularly volcanic islands, can have really incorrect origin estimates, then using such estimates as calibration points is flawed. But if the estimates were truly too old, we would expect to see much older divergence time estimates in each group, which we do not.
I agree that phylogenomics would be useful for addressing these questions. But, the authors should provide a stronger case for why RAD-seq is the way forward rather than other phylogenomic methods.
There is very little discussion of plants in this review. The authors should take that out of the abstract or provide more details.
Finally, some moderate English edits are necessary.
Author Response
REVIEWER#2:
R2/Q1: Comments and Suggestions for Authors
Review of Manuscript ijms-465936 “Shortcomings of phylogenetic studies on recent radiated insular groups: a meta-analysis using Cabo Verde biodiversity”:
The manuscript by Romeiras et al. is, according to the authors, a review of Macaronesian island phylogenetic studies of plants and reptiles and the shortcomings of those methods for studying insular groups. The authors present a description and review of phylogenetic methods in some reptile genera and provide a list of the markers, methods, and programs used for phylogenetic analysis and divergence time estimation in those papers. The authors then attempt to compare studies of reptile phylogeny in the Macaronesian islands by presenting data on the number of studied taxa in a given group, the number of calibrations for divergence time estimation, and the number of unresolved clades for each genus. They then discuss the relative divergence time estimates for different genera among studies and present one or two possible reasons for these differences. Finally, the authors recommend phylogenomics (and RAD-seq in particular) for resolving such issues.
There are some potentially valuable components to both the goals and implementation of this manuscript. First, it is reasonable to question whether past or current phylogenetic methods have been able to resolve phylogenetic questions in any group, and this question is particularly important when thinking about recently radiated lineages due to possible incomplete lineage sorting or hybridization, among other things. Additionally, Table 1 provides very relevant information for any authors interested in pursuing phylogenetic research on these groups, as it is a succinct and thorough list of information from each study. However, I see substantial problems with this manuscript as written, and I detail some of my main points below.
RESPONSE: We thank the Reviewer#2 for his helpful feedback and very useful comments - we have responded to the specific points below.
R2/Q2: First and foremost, the authors do not provide satisfying information that directly addresses their stated goal of discussing the adequacy of the given methods for the target groups. The beginning of the manuscript focuses on the different steps in phylogenetic inference, and details alignment and phylogeny inference methods (and the software programs one can use), but only mentions some standard caveats to each method, with no real novel insight into how these caveats are impacting their groups of interest. Furthermore, much of this information, including lists of programs that are available for alignment and phylogeny inference, can be found in any number of reviews or book chapters on the subject.
RESPONSE: We have carefully considered this comment by the Reviewer and so we have profoundly changed the section ”2. Phylogenetic inference and divergence time estimation”. In particular, the literature on phylogenetic studies with plant lineages (see Lines: 73-80; 90-94; 106-113; and 136-149) have been reviewed and details have been added for species found in Cabo Verde and in Macaronesian region in general. Also, we have amended the aims to be clearer that reptiles and plants are good model systems to understand patterns of diversification within this archipelago (Lines 58-64).
Moreover, and following the Reviewer suggestions, we have removed several paragraphs that are easily available in published reviews/book chapters on the subject. Below we indicated the paragraphs that were removed from the revised version of the manuscript:
“Sequence alignment is an initial and fundamental step to any phylogenetic inference study. A considerable variety of alignment software with specifically-developed algorithms and diverse functionalities is currently available for processing sequence data, and several of these programs have been used in Macaronesian studies. Examples of these programs are MAFFT [9], BLAST [10], MUSCLE [11] and Clustal [12], which have undergone countless improvements over time and are publicly available online and through other software applications as a built-in option.” (Former Lines 69-75).
“Maximum Parsimony (MP) [13-14] infers a phylogenetic tree involving the identification of a tree topology that requires the smallest number of character-state changes to explain a given dataset and is usually well-justified statistically if the number of site changes over evolutionary times and the potential fraction of characters showing homoplasy, is small.” (Former Lines 78-82).
“A method known as the Markov Chain Monte Carlo (MCMC), which reduces computational demands drastically, is then used to sample the most probable trees from a distribution of posterior probabilities.” (Former Lines 95-97).
“Used as a standard for ML analysis, RAxML adapts the algorithm developed by [54], which was the first to compute phylogenetic trees given a set of DNA data and works by selecting the tree that has the highest probability of generating the observed data. Several optimizations and updates were built-in the program, such as “rapid bootstrap” [52], which is extremely fast and capable of analysing big volumes of data.” (Former Lines 108-112).
“To conduct a solid divergence time estimation, the selection of an appropriate clock model is an important step. The simplest approach, a strict-clock, assumes that the evolution rates are fixed along the branches, and even thought it performs well when low rate variations are expected, it is usually unrealistic and not adequate for most of the analysis [57]. To correct this issue, relaxed clock methods were introduced, such as those using autocorrelated substitution rates across branches, uncorrelated substitution rates across branches and random local molecular clocks [58].” (Former Lines 118-124)
“Proper visualization and manipulation of inferred phylogenetic trees and chronograms is essential, and several software has been used in studies with the target Macaronesian lineages. Among these are FigTree [66], TreeGraph2 [67], Dendroscope3 [68] and iTOL3 [69], which range in both complexity and offered tools (references provided in Table 1).” (Former Lines 147-150).
Finally, the literature has been carefully reviewed in order to clarify the discussion of the phylogenetic methods used for Cabo Verde endemic plant lineages, and new references were added to the revised version of the ms and are highlighted in blue in the References section.
Also, on Table 1 we have added one new column (Year) providing exhaustive information and a thorough review of the studies performed with reptiles.
R2/Q3: As a reader, I expected a more thorough look at the phylogenies generated using these methods, and a discussion of how they compared among studies that used similar or different methodologies. The authors attempted to do this in section 4 (starting on line 187), but their analyses were underwhelming. They provide no information on the overall consensus of phylogenies generated in the studies they highlight, with the exception of Figure 3 where the number of unresolved clades is presented for each group.
RESPONSE: We agree with the Reviewer that several aspects were not sufficiently explained in the first version of our ms, therefore we have tried to be more precise in this new version. Specifically, in the section 4, we improved the descriptions of the selected studies for each group, by developing a more extensive discussion on their methodology and main results. Each reptile genus has its own unique research history, with different struggles and successes, and we added this information to the updated text as well. We believe that the new revised section covers the main issue pointed out by the reviewer (Lines 282-318).
R2/Q4: However, not only do I not know which y-axis refers to calibration points v. number of unresolved clades, I do not know how the authors are determining what is an “unresolved clade”. What does that mean in the context of their analyses? And how many clades are present in each of those groups? The authors also do not discuss context for each of these groups or discuss how the different evolutionary history in the three genera may impact phylogeny inference.
RESPONSE: We thank the reviewer for highlighting this point. We have rearranged the Figures 2, 3 and 4 and we have also added more complete legends to improve the explanations of these figures. Specifically, all graphic figures were corrected and expanded upon, by adding more information. In Figure 2, axis are now included, as well as minor description in its legend that point to each analysis. The same was done for Figure 3, in addition to the number of unresolved clades by the total number of clades, which was added to each vertical bar. Relatively to what we determined as an unresolved clade, we based its classification on the obtained support, and as such a clade was considered unresolved when insufficient support was found, i.e. under 70% for ML and 0.90 for BI (based on Hillis, D.M., & Bull, J.J. An empirical test of bootstrapping as a method for assessing confidence in phylogenetic analysis. Systematic biology 1993, 42(2), 182-192)
R2/Q5: The meta-analysis results on divergence time are interesting, but somewhat confusing. It is clear that the type of calibration seems to matter (in some genera more than others), but there is no comparison between geologic dating in insular groups compared to dating in non-insular groups. Since their argument is that islands, particularly volcanic islands, can have really incorrect origin estimates, then using such estimates as calibration points is flawed. But if the estimates were truly too old, we would expect to see much older divergence time estimates in each group, which we do not.
RESPONSE: We understand the reviewer’s viewpoint here, and we do agree that a brief overview to compare geologic dating in insular groups with dating in non-insular groups could be advantageous to provide more context of our discussion (Lines 261-267). However we would like to clarify that the use of geological dating in non-insular is very sparse and restricted to very particular regions and/or geological formations. As we further develop in the text, the same rules and cares should be applied when using these times of calibrations. We added a short description of this topic in section 4 that comprises this subject, both generally speaking and in a particular case (Qinghai-Tibetan Plateau).
R2/Q6: I agree that phylogenomics would be useful for addressing these questions. But, the authors should provide a stronger case for why RAD-seq is the way forward rather than other phylogenomic methods.
RESPONSE: We feel that it is important and relevant to at least briefly highlight that there is other phylogenomic methods can be used to address on recent radiated species. We have restructuring the final sentence of the section 5 providing additional context for this particular point (Lines 369-372). This also provides context for further discussion of other phylogenomics methods, namely High Throughput Sequencing (HTS) techniques, which can provide the opportunity to use thousands of loci to address phylogenetic problems with recent radiated lineages (Lines 399-405). Also a final sentence was added to the final remarks (Lines 427-429).
R2/Q7: There is very little discussion of plants in this review. The authors should take that out of the abstract or provide more details.
RESPONSE: We have carefully considered this comment by the Reviewer, and so we have profoundly changed the manuscript. As stated above, in the revised version of the manuscript new information is provided, particularly in the section 2 (see Lines 73-80; 90-94; 106-113; and 136-149). Also at the end of the section 4 more details are now provided (Lines 189-192; and 321-327) and the section 5 was improved to better envisage that Phylogenomic methods can also be used with the Macaronesian lineages as model to study evolutionary processes of genotype-to-phenotype adaptations, and explore the adaptation of different taxa to certain habitat conditions, within each archipelago (Lines 369-372). Nevertheless, we change some words in the abstract (Line 25) and the Keywords were changed “reptiles; flora” to “endemic species”.
R2/Q8: Finally, some moderate English edits are necessary.
RESPONSE: We would like to clarify that the language in the manuscript has been extensively checked and revised so as to improve syntax and the text’s general fluidity.
We hope that these improvements have adequately addressed all the Reviewers concerns.
Yours sincerely,
Maria Romeiras